# Association of High-Sensitivity C-Reactive Protein and Alcohol Consumption on Metabolic Syndrome in Korean Men

**DOI:** 10.3390/ijerph19052571

**Published:** 2022-02-23

**Authors:** Yong Woo Lee, Sung Soo Kim, Won Yoon Suh, Yu Ri Seo, Sami Lee, Hyun Gu Kim, Jong Sung Kim, Seok Jun Yoon, Jin Gyu Jung

**Affiliations:** 1Department of Family Medicine, Chungnam National University Hospital, Daejeon 35015, Korea; tartspirit@naver.com (Y.W.L.); babydrkim@cnuh.co.kr (H.G.K.); 2Department of Family Medicine, Chungnam National University Hospital, Chungnam National University College of Medicine, Daejeon 35015, Korea; greensea@cnuh.co.kr (W.Y.S.); josephkim@cnu.ac.kr (J.S.K.); sj0219@cnuh.co.kr (S.J.Y.); jjg72@cnuh.co.kr (J.G.J.); 3Department of Family Medicine, Sejong Chungnam National University Hospital, Sejong 30099, Korea; seoglass@cnuh.co.kr (Y.R.S.); smlee@cnuh.co.kr (S.L.)

**Keywords:** C-reactive protein, alcohol, metabolic syndrome, biomarker, health promotion

## Abstract

Background: This study aimed to examine the effect of both alcohol consumption and high-sensitivity C-reactive protein (hsCRP) on metabolic syndrome (MetS) in Korean men. Methods: A cohort of 364 men included in this study was divided into four groups according to the amount of alcohol they consumed: the nondrinkers (ND), low moderate drinkers (LM, ≤7 standard drinks per week), high moderate drinkers (HM, 7 to 14 drinks per week), and heavy drinkers (HD, >14 drinks per week). Logistic regression analyses were performed after adjusting for age, exercise, and smoking. Results: The risk of MetS in the LM group with a high hsCRP level (1.0 or more mg/dL) was not significant. However, the risks of MetS were significantly higher in the HM and HD groups with high hsCRP levels than that in the ND group. The odds ratios of MetS in the HM and HD groups with high hsCRP levels were 3.44 (95% confidence interval (CI), 1.25–9.52) and 3.14 (95% CI, 1.07–9.23), respectively. Conclusion: This study suggests that the risk of MetS is higher in men who consume more than seven drinks a week with high hsCRP levels than that in nondrinkers.

## 1. Introduction

Metabolic syndrome is a major risk factor of type II diabetes and cardiovascular disease [1,2]. Studies on the association between alcohol consumption and metabolic syndrome showed highly diverse results. Alcohol consumption is one of the factors that affect metabolic syndrome. Most studies assessing the association between alcohol consumption and metabolic syndrome showed positive association between heavy drinking and metabolic syndrome [3,4,5,6]. On the contrary, the association between metabolic syndrome and alcohol consumption was shown to be diverse in case of moderate drinking [5,6,7].

A high-sensitivity C-reactive protein (hsCRP) can measure very low concentration using CRP-specific antigen [8]. It has been known that hsCRP is associated with infection, trauma, infarction, inflammatory arthritis, and other disorders including autoimmune disease, inflammatory disease, and various neoplasms. Moreover, the hsCRP level increases in patients with cardiovascular diseases [9]. In 2019, the American College of Cardiology/American Heart Association on the primary prevention of cardiovascular disease suggested the hsCRP level as a risk enhancer of atherosclerotic cardiovascular disease [9]. Moreover, it has been revealed that associations between various cardiovascular diseases and hsCRP exist [10]. An elevated level of hsCRP has been shown to be associated with diseases that could be explained by chronic inflammatory states such as asymptomatic cerebral infarction, diabetes, and non-alcoholic fatty liver disease [11,12,13]. Metabolic syndrome is accompanied by chronic inflammatory responses [14] and usually has a positive association with hsCRP [15,16].

Additionally, Xu et al. [17] suggested that the risk of increased hsCRP is higher as the alcohol consumption increases. Most studies on the effect of alcohol on the metabolic syndrome according to the hsCRP level were evaluated by correcting alcohol as a confounding factor or classifying it into nondrinkers, moderate drinkers, and heavy drinkers. Few studies have simultaneously evaluated the effects of alcohol consumption and hsCRP on metabolic syndrome. The moderate amount of alcohol that is commonly known (standard drink of 14 or less drinks per week) may be lower for Koreans who are more sensitive to alcohol than the West [18].

Thus, the present study investigated the effects of both alcohol consumption and hsCRP on metabolic syndrome after the moderate-drinking group was subdivided.

## 2. Materials and Methods

### 2.1. Subjects

Initially, 373 men who underwent the hsCRP test in a university hospital health promotion center from October 2016 to March 2017 were considered as study subjects. Considering that the rate of heavy drinkers in women is low, women were excluded. Finally, 364 patients were selected as study subjects after, excluding 9 men who had trauma (0), which could induce an increased hsCRP level, within 3–6 months, who had acute myocardial infarction (0), who were diagnosed with infectious diseases (8), and who were diagnosed/treated for rheumatoid arthritis or connective tissue diseases (1).

### 2.2. Study Methods

The present study was designed as a cross-sectional study. Data on alcohol consumption, smoking status, exercise, medical history, and diseases currently being treated were collected by analysis using a self-administered questionnaire. Heights were measured by an automatic height meter in 0.1 cm unit. Weights were measured in 0.01 kg units, with the subject wearing a patient’s gown. Body mass index (BMI) was calculated by dividing weight (kg) by height (m) squared. Waist circumference (WC) was measured on the direct upper level of the iliac crest in 0.1 cm unit at the end of exhalation using a tape measure in parallel with the floor. Blood pressure (BP) was measured at the forearm with the subject in a resting state using an automated BP device after stabilization for approximately 10 min. Each subject’s blood was drawn after 12 h of fasting to accurately measure the triglyceride (TG), high-density lipoprotein (HDL), and fasting plasma glucose (FPG) levels, which are laboratory components of metabolic syndrome.

For the smoking status, subjects were divided into non-smokers, ex-smokers, and current smokers. Exercise groups were divided into a group without exercise, a group having irregular exercise for more than 30 min less than 3 times per week, and a group having regular exercise for more than 30 min at least 3 times per week.

For alcohol consumption-related characteristics, a questionnaire was used to evaluate the amount of alcohol consumption (drinks or bottles) and the number of weekly alcohol consumption. Following the criteria of the National Institute on Alcohol Abuse and Alcoholism (NIAAA), 1 standard drink (hereafter drink) was defined as an amount containing 14 g of alcohol [19]. According to the standard of the NIAAA, drinkers were divided into heavy drinkers (HD) with more than 14 drinks per week and moderate drinkers with 14 drinks or less per week [19]. Moderate drinkers were further subdivided into low moderate drinkers (LM) with 7 drinks or less per week and high moderate drinkers (HM) with more than 7 drinks and not more than 14 drinks per week. On the contrary, the hsCRP level was classified into a high hsCRP (≥1.0 mg/dL) and low hsCRP (<1.0 mg/dL) level based on the low-risk cut-off level of hsCRP (1.0 mg/dL) as recommended by the Center for Disease Control and the American Heart Association [20].

Metabolic syndrome was defined as a state that met at least three conditions of metabolic syndrome following the diagnostic criteria of the National Cholesterol Education Program Adult Treatment Panel III. In detail, the diagnostic criteria of metabolic syndrome were as follows [21]: WC ≥ 90 cm, TG ≥ 150 mg/dL, HDL ≤ 40 mg/dL, systolic BP ≥ 130 mmHg or diastolic BP ≥ 85 mmHg, and FPG ≥ 100 mg/dL. In case of taking medication for hypertension, diabetes, and dyslipidemia, each of them was accounted for the respective condition of metabolic syndrome. The standard for WC of abdominal obesity was based on the guideline of the Korean Society for the Study of Obesity [22].

### 2.3. Statistical Methods

Continuous categorical variables were analyzed using the analysis of variance and *t*-test, whereas categorical variables were analyzed using the chi-squared test. Post hoc analysis was performed by the Scheffe test. A variable (hsCRP) that does not have a normal distribution was analyzed by Kruskal–Wallis H test and Mann-Whitney U test. Additionally, logistic regression analysis was performed to evaluate the risk for metabolic syndrome after adjusting for age, exercise, and smoking status. The statistically significant level was set to *p* = 0.05, and all other statistical analyses were performed using IBM SPSS Statistics version 26 (IBM Corporation, Armonk, NY, USA).

## 3. Results

### 3.1. Characteristics of Subjects According to Alcohol Consumption

There was a total of 364 subjects including 55 NDs (15.2%), 179 LM drinkers (49.3%), 68 HM drinkers (18.7%), and 62 HDs (17.1%). Their mean weekly alcohol consumption was 7.0 ± 8.5 drinks, including 0 drinks for NDs, 2.8 ± 1.8 drinks for LM drinkers, 9.6 ± 1.9 drinks for HM drinkers, and 22.4 ± 8.4 drinks for HDs. Regarding age, The HD group was younger than the ND, LM, and HM groups. BMI, WC, systolic blood pressure (SBP), diastolic blood pressure (DBP), FPG, and HDL were not significantly different between the four groups. The TG was significantly higher in the HD group than in the ND and LM groups. There was no significant difference in hsCRP between the four groups. The HD group (54%) had the highest rate of smokers, followed by the HM (39.7%), LM (28.5%), and ND (20%) groups in order (Table 1).

### 3.2. Characteristics of Subjects with or without Metabolic Syndrome

Age, exercise, and smoking status were not significantly different between the two groups. The total weekly alcohol consumption was significantly higher in the group with metabolic syndrome, but binge and alcohol consumption of LM, HM, and HD were not significant between the two groups. BMI, WC, SBP, DBP, FPG, HDL, TG, and hsCRP were significantly higher in the group with metabolic syndrome (Table 2).

### 3.3. Characteristics of Subjects According to hsCRP Levels

Weekly alcohol consumptions were 6.8 ± 8.8 drinks for the low hsCRP group and 7.5 ± 7.7 drinks for the high hsCRP group, showing no significant difference. The high hsCRP group had significantly higher BMI, WC, and FPG levels and a significantly lower HDL level than the low hsCRP group. The proportions of regular exercise at least three times per week were 37.6% in the low hsCRP group and 35.8% in the high hsCRP group, showing no significant difference. The proportions of smokers were 34.1% in the low hsCRP group and 33.0% in the high hsCRP group, with no significant difference between the two groups (Table 3).

### 3.4. Odds Ratios of Metabolic Syndrome According to Alcohol Consumption and hsCRP Levels

In the total drinkers, after adjusting for age, smoking, and exercise, only the OR of HD group (2.37, *p* = 0.034) for metabolic syndrome was significantly higher than that of the ND group. In low hsCRP (<1.0 mg/dL), the ORs of all groups (LM, HM and HD) were not significant compared to that of the ND group. In the high hsCRP (≥1.0 mg/dL), the ORs of the HM and HD groups were significantly higher than that of the ND group, and were 3.44 (95% CI, 1.25–9.52; *p* = 0.017) and 3.14 (95% CI, 1.07–9.23; *p* = 0.038), respectively (Table 4).

### 3.5. Characteristics of Components According to Alcohol Consumption in MetS with High hsCRP

Regarding metabolic syndrome with high hsCRP (≥1.0 mg/dL), there were no significant differences in each component between the four groups (nondrinkers, LM, HM, HD) according to the amount of alcohol consumed. Additionally, there was no significant difference in the prevalence of metabolic syndrome (Table 5).

## 4. Discussion

The present study aimed to evaluate the effect of both alcohol consumption and hsCRP on metabolic syndrome. Studies on the induction of metabolic syndrome by alcohol consumption showed diverse results, which were attributable to different criteria for alcohol consumption, the degree of severe alcohol-use disorder, type of alcohol consumption for each patient, and genetic factors for alcohol metabolism depending on the study. The association between alcohol consumption and metabolic syndrome depending on the level of hsCRP, a marker of chronic inflammation, among such relevant factors, has been rarely studied.

Most studies assessing the association between alcohol consumption and metabolic syndrome showed a positive association between heavy drinking and metabolic syndrome. In a study comprising Koreans, Oh et al. [3] reported that the risk for metabolic syndrome was higher in the frequent alcohol-consumption group with at least 4 drinks per week and the heavy drinker group with at least 10 drinks on one occasion than that in the nondrinkers. Baik et al.’s prospective study [4] showed that the risk for metabolic syndrome in the heavy-drinking group with greater than 30 g of alcohol consumption per day was higher than that in nondrinkers. In a cross-sectional study comprising Japanese men, Wakabayashi et al. [5] found that the risk for metabolic syndrome in the heavy-drinking group with 44 g of alcohol consumption per day was higher than that in the ND group. Hirakawa et al. [6] showed that the risk for metabolic syndrome in the heavy-drinking group with 60 g of alcohol consumption per day was higher than that in ND group. The present study also found that the risk for metabolic syndrome in the heavy-drinking groups with greater than 14 drinks (28 g/day) per week was higher than that in the ND group (OR, 2.37). Such results were consistent with the results of other studies that showed higher risks for metabolic syndrome in the heavy-drinking groups.

Moderate alcohol consumption has various effects on metabolic syndrome. In a study comprising Korean adults, Kim et al. [7] found that the risk for metabolic syndrome in the moderate alcohol-consumption group (0.1–5.0 g/day) was lower than that in the ND group. Yoon et al. [8] also reported that the moderate alcohol-consumption group with 1–14.9 g/day had a lower risk for metabolic syndrome than the ND group. In the Wakabayashi et al.’s cross-sectional study comprising Japanese men [5], the moderate alcohol-consumption group consuming less than 22 g of alcohol per day had a lower risk for metabolic syndrome than the ND group. Hirakawa et al. [6] also demonstrated that the risk for metabolic syndrome in the moderate alcohol-consumption group consuming 20 g per day of alcohol was lower than that in the ND group. However, Oh’s study [23] found no significant difference in the risk for metabolic syndrome when the moderate-drinking group consuming 12 or less drinks per week was compared with the ND group. Baik et al.’s prospective study [4] found no significant difference in the relative risk for metabolic syndrome when 0.1–5.0 g/day of alcohol consumption corresponding to the moderate alcohol-consumption group, 5.1–15 g/day of alcohol consumption, and 15.1–30 g/day of alcohol consumption were compared with the ND group.

In the present study, there was no significant difference in the risk for metabolic syndrome between the moderate alcohol-consumption group and the ND group, which was consistent with the results of previous studies by Oh [23] and Baik et al. [4]. Although there was no increase in the risk for metabolic syndrome, the moderate-drinking groups showed either a lower risk or no significant difference depending on the study, which may be explained by the effects of various confounding factors such as meals, activities, and alcohol metabolism.

There is a positive association between hsCRP and metabolic syndrome. In a study using the 2015′s Korean Health and Nutrition Examination survey, Kim et al. [15] found an association between each condition of metabolic syndrome and increased hsCRP level and also showed a higher hsCRP level in a group with metabolic syndrome. Kawamoto et al. [16] reported that a higher hsCRP level was synergistically associated with metabolic syndrome and insulin resistance. In our study, the alcohol-consumption groups (the HM and HD groups) with at least seven drinks per week showed an increased risk for metabolic syndrome under a high hsCRP level, as in the above results, whereas the group with a low hsCRP level had no significant increase in the risk (Table 3).

Oliveira et al. [24] reported a linear dose–response relationship between alcohol consumption and hsCRP in men. In our study, the hsCRP level showed a high tendency in the heavy-drinking group than in the other groups, but there was no significant difference. In the study by Oliveira et al. [24], the daily alcohol consumption of the heavy-drinking group (>30 g/d) was 203 g, which was 4.5 times higher than that of the heavy-drinkers group in our study, 44.8 g (22.4 standard drinks/week). The reason that the hsCRP was not significantly high in the heavy-drinkers group in our study could be explained by the fact that the absolute amount of alcohol consumed was low. Additionally, the risk for metabolic syndrome in the heavy-drinkers group with a high hsCRP level (OR, 3.14) was higher than that of the total heavy-drinkers groups (OR, 2.37). Moreover, the HM (7–14 standard drinks/week) group with a high hsCRP level had an increased risk for metabolic syndrome (OR, 3.44), although they were in the moderate alcohol-consumption group. Such results were different from the results of general studies that moderate alcohol-consumption groups had no increase in the risk for metabolic syndrome.

In general, alcohol is converted to acetaldehyde in the liver by alcohol dehydrogenase (ADH) in the cytosol, microsomal cytochrome p450 (CYP) 2E1, and microsomal catalase of peroxisome. In case of a low amount of alcohol consumption, ADH degrades alcohol, whereas CYP 2E1 and catalase degrade alcohol in case of heavy alcohol drinking. Hence, in case of low levels of ADH metabolism and CYP2E1 metabolic activity or heavy drinking, oxidative stress is caused by antioxidant deficiency and the generation of reactive oxygen species [25,26].

In this study, if hsCRP is increased due to influences other than alcohol consumption, the risk of metabolic syndrome may increase, even if it is lower than the moderate consumption. In high hsCRP, the risk of metabolic syndrome tends to increase with alcohol consumption. However, the odds ratios of HM and HD are similar. In metabolic syndrome with high hsCRP, there was no significant difference in the components and prevalence of metabolic syndrome according to the amount of alcohol consumed (Table 5). These results suggest that, in the case of high hsCRP, the effect of alcohol on metabolic syndrome may not be dose-dependent. Therefore, in the case of high hsCRP, it is considered that at least the oxidative stress caused by drinking may be stronger.

Xu et al. [17] reported that alcohol sensitivity (facial flushing, palpitation, or dizziness after drinking) should be associated with an elevated hsCRP level in alcohol drinkers. Such alcohol-induced flushing is often observed in East Asians, which is related to the polymorphism of enzymes in the pathway of alcohol metabolism [27,28]. Enzyme deficiency caused by mutations in aldehyde dehydrogenase 2 (ALDH2) gene interferes with alcohol metabolism and induces unpleasant feelings, palpitations, and headache after alcohol drinking and alcohol-induced flushing due to the accumulation of acetaldehyde in the body [29]. In a study comprising Koreans, Kim et al. [18] reported that the group with facial flushing indicating alcohol sensitivity had a higher risk for metabolic syndrome related to alcohol consumption than the group without it. Therefore, in-drinking groups, the higher risk for metabolic syndrome in the high hsCRP group than that of the low hsCRP group can be explained by such alcohol sensitivity, although the specific reason remains to be elucidated.

This study has the following limitations: first, this study was unable to explain the causal relationship between alcohol consumption and metabolic syndrome based on the hsCRP level due to its cross-sectional study design. Second, this study was not able to represent the general characteristics of Korean men because it targeted subjects who underwent a health checkup. Third, women were excluded in this study because of the low proportion of alcohol-drinking women. Thus, large-scale prospective studies should be conducted in the future.

## 5. Conclusions

If hsCRP is high, the risk of metabolic syndrome is significantly increased in the groups (HM and HD) drinking more than seven drinks per week. The risk of metabolic syndrome shows a tendency to increase, although it was not significant in the LM group (seven drinks or less per week) when hsCRP is high. We suggest that if hsCRP is high, abstaining from alcohol may be helpful to reduce the risk of metabolic syndrome.

## Figures and Tables

**Table 1 ijerph-19-02571-t001:** Characteristics of subjects according to alcohol consumption.

Variable	Nondrinkers (*n* = 55)	Drinkers (*n* = 309)			*p*-Value *
		LM (*n* = 179)	HM (*n* = 68)	HD (*n* = 62)	
Age (year)	48.6 ± 8.3 ^a^	45.8 ± 8.7 ^a^	45.1 ± 7.8 ^a^	44.2 ± 7.4 ^b^	0.028
BMI (kg/m^2^)	24.8 ± 3.8 ^a^	24.9 ± 3.1 ^a^	25.6 ± 3.1 ^a^	25.8 ± 3.0 ^a^	0.089
WC (cm)	85.6 ± 10.1 ^a^	87.4 ± 8.9 ^a^	88.9 ± 8.1 ^a^	90.5 ± 7.9 ^a^	0.047
SBP (mmHg)	125.6 ± 13.4 ^a^	125.8 ± 14.3 ^a^	126.5 ± 12.2 ^a^	128.4 ± 14.6 ^a^	0.598
DBP (mmHg)	77.9 ± 9.2 ^a^	79.4 ± 10.6 ^a^	81.4 ± 8.3 ^a^	82.1 ± 10.0 ^a^	0.061
FPG (mg/dL)	101.6 ± 17.3 ^a^	101.3 ± 22.7 ^a^	103.1 ± 18.0 ^a^	101.6 ± 19.7 ^a^	0.945
HDL (mg/dL)	51.5 ± 13.2 ^a^	51.1 ± 11.3 ^a^	50.8 ± 10.3 ^a^	53.9 ± 13.4 ^a^	0.385
TG (mg/dL)	129.5 ± 66.3 ^a^	142.7 ± 107.5 ^a^	167.5 ± 84.9 ^a,b^	191.0 ± 117.9 ^b^	0.002
hsCRP (mg/L)	0.50 (0.3–0.9)	0.50 (0.3–1.0)	0.55 (0.3–1.2)	0.70 (0.3–1.3)	0.313
Exercise					0.289
No	9 (16.4)	37 (20.7)	17 (25.0)	18 (29.0)	
Irregular	20 (36.4)	75 (41.9)	32 (47.1)	21 (33.9)	
Regular	26 (47.3)	67 (37.4)	19 (27.9)	23 (37.1)	
Smoking status					<0.001
Non-smoker	29 (52.7)	56 (31.3)	21 (30.9)	11 (17.7)	
Ex-smoker	15 (27.3)	72 (40.2)	20 (29.4)	17 (27.4)	
Current smoker	11 (20.0)	51 (28.5)	27 (39.7)	34 (54.8)	
Drink/week ^†^	0	2.8 ± 1.8 ^a^	9.6 ± 1.9 ^b^	22.4 ± 8.4 ^c^	<0.001
Binge ^†^ Prevalance of MetS (%)	0 27.3 (15/55) ^a,b^	6.9 ± 5.1 ^a^ 24.6 (44/179) ^b^	10.2 ± 4.2 ^b^ 36.8 (25/68) ^a,b^	12.2 ± 4.7 ^b^ 43.5 (27/62) ^a^	<0.001 0.024

Values are expressed as mean ± standard deviation, number (%) or median (interquartile range). * *p* values are obtained by analysis of variance for continuous variables, by chi-squared test for categorical variables and by Kruskal–Wallis H test for hsCRP. ^a,b,c^ The same letters indicate non-significant difference between groups based on post hoc test (Scheffe for continuous variables, Bonferroni for categorical variable). ^†^ One standard drink = 14 g of alcohol; Binge, 5 or more drinks in a 2 h time frame; LM, low moderate drinkers (≤7 drinks per week); HM, high moderate drinkers (7 to 14 drinks per week); HD, heavy drinkers (>14 drinks per week). No exercise is zero exercise per week, irregular exercise is less than 3 times per week, and regular exercise is 3 times or more per week. BMI, body mass index; WC, waist circumference; SBP, systolic blood pressure; DBP, diastolic blood pressure; FPG, fasting plasma glucose; HDL, high-density lipoprotein; TG, triglyceride; hsCRP, high-sensitivity C-reactive protein.

**Table 2 ijerph-19-02571-t002:** Characteristics of subjects with or without metabolic syndrome.

Variable	Metabolic Syndrome		*p*-Value *
	Absence (*n* = 253)	Presence (*n* = 111)	
Age (year)	45.4 ± 8.2	46.8 ± 8.6	0.135
BMI (kg/m^2^)	24.1 ± 2.6	27.6 ± 3.1	<0.001
WC (cm)	85.0 ± 7.4	95.1 ± 8.0	<0.001
SBP (mmHg)	124.6 ± 13.4	128.1 ± 14.8	<0.001
DBP (mmHg)	79.5 ± 9.7	81.2 ± 10.4	<0.001
FPG (mg/dL)	99.3 ± 16.8	107.4 ± 26.7	<0.001
HDL (mg/dL)	52.7 ± 12.4	49.0 ± 10.0	<0.001
TG (mg/dL) hsCRP (mg/L)	148.0 ± 98.0 0.4 (0.0–0.9)	166.6 ± 110.3 0.9 (0.4–1.5)	<0.001 <0.001
Exercise			0.436
No	56 (22.1)	25 (22.5)	
Irregular	108 (42.7)	40 (36.0)	
Regular	89 (35.2)	46 (41.4)	
Smoking status			0.167
Non-smoker	87(34.4)	30(27.0)	
Ex-smoker	88(34.8)	36(32.4)	
Current smoker	78(30.8)	45(40.5)	
Drink/week ^†^	6.2 ± 7.8	8.7 ± 9.6	0.020
LM	2.8 ± 1.7	2.8 ± 2.0	0.782
HM	9.5 ± 1.8	9.7 ± 2.0	0.716
HD	22.5 ± 7.9	22.2 ± 9.2	0.901
Binge ^†^	7.2 ± 6.1	7.9 ± 5.0	0.901

Values are expressed as mean ± standard deviation, number (%) or median (interquartile range). * *p* values are obtained by analysis of variance for continuous variables, by chi-squared test for categorical variables and by Mann-Whitney U test for hsCRP. ^†^ One standard drink = 14 g of alcohol; LM, low moderate drinkers (≤7 drinks per week); HM, high moderate drinkers (7 to 14 drinks per week); HD, heavy drinkers (>14 drinks per week); Binge, 5 or more drinks in a 2 h time frame. No exercise is zero exercise per week, irregular exercise is less than 3 times per week, and regular exercise is 3 times or more per week. BMI, body mass index; WC, waist circumference; SBP, systolic blood pressure; DBP, diastolic blood pressure; FPG, fasting plasma glucose; HDL, high-density lipoprotein; TG, triglyceride; hsCRP, high-sensitivity C-reactive protein.

**Table 3 ijerph-19-02571-t003:** Characteristics of subjects according to hsCRP levels.

Variable	Cutoff Point = 1.0 mg/dL		*p*-Value *
	Low hsCRP (*n* = 255)	High hsCRP (*n* = 109)	
Age (year)	45.9 ± 8.2	45.5 ± 8.8	0.899
BMI (kg/m^2^)	24.7 ± 2.8	26.2 ± 3.7	0.001
WC (cm)	86.7 ± 7.9	91.4 ± 10.2	0.001
SBP (mmHg)	124.6 ± 13.4	128.1 ± 14.8	0.346
DBP (mmHg)	79.5 ± 9.7	81.2 ± 10.4	0.543
FPG (mg/dL)	99.3 ± 16.8	107.4 ± 26.7	<0.001
HDL (mg/dL)	52.7 ± 12.4	49.0 ± 10.0	0.040
TG (mg/dL)	148.0 ± 98.0	166.6 ± 110.3	0.133
Exercise			0.881
No	55 (21.6)	26 (23.9)	
Irregular	104 (40.8)	44 (40.4)	
Regular	96 (37.6)	39 (35.8)	
Smoking status			0.756
Non-smoker	79(31.0)	38(34.9)	
Ex-smoker	89(34.9)	35(32.1)	
Current smoker	87(34.1)	36(33.0)	
Drink/week ^†^	6.8 ± 8.8	7.5 ± 7.7	0.864
LM	2.8 ± 1.7	2.8 ± 1.9	0.850
HM	9.2 ± 1.7	10.2 ± 1.0	0.037
HD	23.5 ± 9.3	20.1 ± 5.9	0.130
Binge ^†^	7.3 ± 5.7	7.6 ± 5.9	0.901
Prevalance of MetS (%)	24.3 (62/255)	45.0 (49/109)	<0.001

Values are expressed as mean ± standard deviation or number (%). * *p* values are obtained by analysis of variance for continuous variables and by Chi-squared test for categorical variables. ^†^ One standard drink = 14 g of alcohol; LM, low moderate drinkers; HM, high moderate drinkers; HD, heavy drinkers; Binge, 5 or more drinks in a 2 h time frame; No exercise is zero exercise per week, irregular exercise is less than 3 times per week, and regular exercise is 3 times or more per week. hsCRP, high-sensitivity C-reactive protein; low hsCRP, <1.0 mg/dL; high hsCRP, ≥1.0 mg/dL; BMI, body mass index; WC, waist circumference; SBP, systolic blood pressure; DBP, diastolic blood pressure; FPG, fasting plasma glucose; HDL, high-density lipoprotein; TG, triglyceride.

**Table 4 ijerph-19-02571-t004:** Odds ratios of metabolic syndrome according to alcohol consumption and high-sensitivity C-reactive protein levels.

Variable	Total Drinkers (*n* = 309)	Low hsCRP (*n* = 213)	High hsCRP (*n* = 96)
ND (*n* = 55)	1	1	1
LM (*n* = 179)	0.94 (0.47–1.88), *p* = 0.866	0.70 (0.33–1.50), *p* = 0.359	1.59 (0.68–3.70), *p* = 0.286
HM (*n* = 68)	1.75 (0.80–3.84), *p* = 0.163	1.01 (0.40–2.56), *p* = 0.988	3.44 (1.25–9.52), *p* = 0.017
HD (*n* = 62)	2.37 (1.07–5.24), *p* = 0.034	1.63 (0.66–4.05), *p* = 0.291	3.14 (1.07–9.23), *p* = 0.038

Obtained by logistic regression analysis after adjustment for age, smoking, and exercise. *p* values are obtained by logistic regression. hsCRP, high-sensitivity C-reactive protein; Low hsCRP, <1.0 mg/dL; high hsCRP, ≥1.0 mg/dL; ND, nondrinkers; LM, low moderate drinkers; HM, high moderate drinkers; HD, heavy drinkers; hsCRP, high-sensitivity C reactive protein.

**Table 5 ijerph-19-02571-t005:** Characteristics of components according to alcohol consumption in MetS with high hsCRP (≥1.0 mg/dL).

Variable	Nondrinkers	Drinkers (*n* = 96)			*p*-Value *
	(*n* = 13)	LM (*n* = 50)	HM (*n* = 25)	HD (*n* = 21)	
Age (year)	46.23 ± 10.5	45.8 ± 9.9	46.3 ± 7.6	44.3 ± 6.0	0.873
BMI (kg/m^2^)	26.3 ± 4.4	25.5 ± 3.9	27.3 ± 3.6	26.3 ± 2.7	0.304
WC (cm)	89.7 ± 12.2	90.5 ± 11.3	93.1 ± 8.8	92.8 ± 7.3	0.613
SBP (mmHg)	132.9 ± 16.6	126.3 ± 16.7	130.2 ± 10.3	126.6 ± 13.1	0.424
DBP (mmHg)	80.2 ± 10.5	79.4 ± 11.9	83.8 ± 7.4	83.2 ± 8.9	0.249
FPG (mg/dL)	101.7 ± 18.5	107.9 ± 32.6	109.6 ± 22.6	107.1 ± 20.3	0.858
HDL (mg/dL)	49.3 ± 9.3	48.6 ± 9.8	47.5 ± 8.5	51.7 ± 12.4	0.529
TG (mg/dL)	133.3 ± 74.7	152.0 ± 119.9	185.6 ± 105.9	199.6 ± 103.8	0.198
Prevalence of MetS (%)	6 (12.2)	18 (36.7)	14 (28.6)	11 (22.4)	0.344

Values are expressed as mean ± standard deviation or number (%). * *p* values are obtained by analysis of variance for continuous variables, by chi-squared test for prevalence of metabolic syndrome (MetS). All variables of each component of MetS are non-significant difference between groups based on post hoc test (Scheffe for continuous variables, Bonferroni for categorical variable). LM, low moderate drinkers (≤7 drinks per week); HM, high moderate drinkers (7 to 14 drinks per week); HD, heavy drinkers (>14 drinks per week); BMI, body mass index; WC, waist circumference; SBP, systolic blood pressure; DBP, diastolic blood pressure; FPG, fasting plasma glucose; HDL, high-density lipoprotein; TG, triglyceride.

## Data Availability

The data are not publicly available due to their containing information that could compromise the privacy of research participants.

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
