# Peer review of "Association of High-Sensitivity C-Reactive Protein and Alcohol Consumption on Metabolic Syndrome in Korean Men"

_ijerph, 2022, doi:10.3390/ijerph19052571_

Round 1

Reviewer 1 Report

MS: IJERPH_1550705

Title: Association of High-sensitivity C-reactive Protein 2 and Alcohol Consumption on Metabolic Syndrome 3 in Korean Men

Comments for authors

This study aims to evaluate the effect of both alcohol consumption and hsCRP on metabolic syndrome (MS) among Korean Men. The association of alcohol drinking in MS is not new but this study is very comprehensive to see the dose-response effect of alcohol on metabolic syndrome together with hsCRP, a marker of chronic inflammation.

MINOR COMMENTS

Methods

  1. Page 2 Line 86 to 92, the number of each case causing exclusion might be useful.

OVERALL IMPRESSION,

I enjoyed reading this MS. Well designed, analyzed, and interpreted.

Author Response

Reviewer 1

Comments for authors

This study aims to evaluate the effect of both alcohol consumption and hsCRP on metabolic syndrome (MS) among Korean Men. The association of alcohol drinking in MS is not new but this study is very comprehensive to see the dose-response effect of alcohol on metabolic syndrome together with hsCRP, a marker of chronic inflammation.

MINOR COMMENTS 

Methods

  1. Page 2 Line 86 to 92, the number of each case causing exclusion might be useful.

Answer (line 90 to 94)

We added the number of each excluded case as follows: Finally, a total of 364 patients were selected as study subjects after excluding 9 men who had trauma (0), which could induce an increased hsCRP level, within 3–6 months, who had acute myocardial infarction (0), who were diagnosed with infectious diseases (8), and who were diagnosed/treated for rheumatoid arthritis or connective tissue diseases (1).

OVERALL IMPRESSION, 

I enjoyed reading this MS. Well designed, analyzed, and interpreted.

Reviewer 2 Report

Comments for the authors

This interesting manuscript study the putative role of high-sensitivity C-reactive protein (hsCRP) on the effects induced by alcohol consumption. In the last years hdCRP has been identified as a marker of inflammation that predicts incident myocardial infarction, stroke, and sudden cardiac death among healthy individuals. In addition, hsCRP confers additional prognostic value at all levels of cholesterol, severity of metabolic syndrome, and blood pressure, and in those patients with atherosclerosis problems. Indeed, in the last years, the role of hsCRP in alcohol induced effects has been evaluated. Thus, the results showed in this manuscript are of interest to readers studying alcohol consumption effects.

The manuscript is well written and the conclusions are well supported by the data. Just some details hampered the enthusiasm raised by this manuscript:

In Table 1 and 2: what you measured as Binge? NIAAA defines binge drinking as a pattern of drinking that brings blood alcohol concentration to 0.08 grams per deciliter (0.08%) or higher. This typically occurs after a woman consumes 4 drinks or a man consumes 5 drinks in a 2-hour time frame. Please explain?

In Table 2 you report the number of Drink/week in volunteers with low hsCRP and high hsCRP. However, why you do not report in addition the number of drink/week in a disaggregate way, considering four categories: nondrinkers, LM, HM and HD. The data showed in this way could help to understand (and compare) better the results showed in Table 3.

Minor comments:

Line 55-56, a reference must be included at the end of the sentence: A high-sensitivity C-reactive protein (hsCRP) can measure very low 55 concentration using CRP-specific antigen

In the same way, a reference is missing at the end of sentences in Line59 and Line 60.

Table 1: The reported P-value for age (years) probably compare differences between Nondrinkers and HD groups. However, probably you don’t found statistical differences between HD and LM and HM groups. If this last is correct, a superscript letter b is missing in LM and HM groups. Please review.

Line 173: Please change:

… irregular exercise is less than 3 per week,

to

irregular exercise is less than 3 times per week,

or correct appropriately because the sentence is not clear. The same correction is needed in Line 174.

Line 272

A dot is missing after reference [4].

Author Response

Reviewer 2

Comments for the authors

This interesting manuscript study the putative role of high-sensitivity C-reactive protein (hsCRP) on the effects induced by alcohol consumption. In the last years hdCRP has been identified as a marker of inflammation that predicts incident myocardial infarction, stroke, and sudden cardiac death among healthy individuals. In addition, hsCRP confers additional prognostic value at all levels of cholesterol, severity of metabolic syndrome, and blood pressure, and in those patients with atherosclerosis problems. Indeed, in the last years, the role of hsCRP in alcohol induced effects has been evaluated. Thus, the results showed in this manuscript are of interest to readers studying alcohol consumption effects.

The manuscript is well written and the conclusions are well supported by the data. Just some details hampered the enthusiasm raised by this manuscript:

In Table 1 and 2: what you measured as Binge? NIAAA defines binge drinking as a pattern of drinking that brings blood alcohol concentration to 0.08 grams per deciliter (0.08%) or higher. This typically occurs after a woman consumes 4 drinks or a man consumes 5 drinks in a 2-hour time frame. Please explain?

Answer (line 176)

I added definition of binge as follows: Binge, 5 or more drinks in a 2-hour time frame.

In Table 2 you report the number of Drink/week in volunteers with low hsCRP and high hsCRP. However, why you do not report in addition the number of drink/week in a disaggregate way, considering four categories: nondrinkers, LM, HM and HD. The data showed in this way could help to understand (and compare) better the results showed in Table 3.

Answer (Table 3 after revision)

As pointed out, the amount of alcohol consumed was divided into subdivisions in Table 3 (Table 2 before revision) such as:

Low hsCRP: LM 2.8±1.7  HM 9.2±1.7  HD 23.5±9.3

High hsCRP: LM2.8±1.9  HM 10.2±1.0  HD 20.1±5.9

Minor comments:

Line 55-56, a reference must be included at the end of the sentence: A high-sensitivity C-reactive protein (hsCRP) can measure very low 55 concentration using CRP-specific antigen

Answer (line 56-57 after revision)

Line 56-57, The existing reference 8 has been deleted and the requested reference has been added as number 8 such as:

Moutachakkir M, Lamrani Hanchi A, Baraou A, Boukhira A, Chellak S. Immunoanalytical characteristics of C-reactive protein and high sensitivity C-reactive protein. Ann Biol Clin (Paris). 2017 Apr 1;75(2):225-229.

In the same way, a reference is missing at the end of sentences in Line59 and Line 60.

Answer (line 60-61 after revison)

The reference 9 has been added.

Table 1: The reported P-value for age (years) probably compare differences between Nondrinkers and HD groups. However, probably you don’t found statistical differences between HD and LM and HM groups. If this last is correct, a superscript letter b is missing in LM and HM groups. Please review.

Answer

As a result of the post-hoc test (Scheffe), there was no significant difference between nondrinkers, LM, and HM (superscript a, respectively), and HD (superscript b) was significantly different from the other 3 groups. Therefore, it is described as: Regarding age, The HD group was younger than the ND, LM and HM groups (line 158-159). 

Line 173: Please change:

… irregular exercise is less than 3 per week,

to

irregular exercise is less than 3 times per week,

or correct appropriately because the sentence is not clear. The same correction is needed in Line 174.

Line 272

A dot is missing after reference [4].

Answer

Line 173, 174, 272 (179, 180, 338 after revision) I have corrected what you pointed out.

Reviewer 3 Report

Dear sir, 

thank you to select me to review an article Association of High-sensitivity C-reactive Protein and Alcohol Consumption on Metabolic Syndrome in Korean Men, written by Lee YW et al. This is cross-sectional study, 364 men were included into final analysis. Patients were divided into four groups: the nondrinkers (ND), low moderate drinkers (LM, ≤7 standard drinks per week), high moderate drinkers (HM, 7 to 14 drinks per week), and heavy drinkers (HD), >14 drinks per week). The risk of MetS is higher in men who consume more than 7 drinks a week with high hsCRP levels than that in nondrinkers. Study is well designed, but I recommend some changes to improve quality of this research: 

1) Please add approval of Ethic committee and data about Inform consent. 

2) Table 1 and table 2: please add rows with data about occurence of MetS, and data about occurence of MetS with 3, 4 and 5 individual components of MetS.

3) Please add  the separate table comparing patients with and without MetS.

4) Please add the separate table comparing only men with high hsCRP, add 4 collumns ND, LM, HM, HD, please include data about occurence of MetS, and data about occurence of MetS with 3, 4 and 5 individual components of MetS.

5) Please add separate table with data from univariate analysis.

6) Part Discussion: Please add chapter with data about association about alcohol and subclinical inflammation

7) Conclusion: Change term individuals for men.

My final decision is major revision. 

Author Response

Reviewer 3

Dear sir, 

thank you to select me to review an article Association of High-sensitivity C-reactive Protein and Alcohol Consumption on Metabolic Syndrome in Korean Men, written by Lee YW et al. This is cross-sectional study, 364 men were included into final analysis. Patients were divided into four groups: the nondrinkers (ND), low moderate drinkers (LM, ≤7 standard drinks per week), high moderate drinkers (HM, 7 to 14 drinks per week), and heavy drinkers (HD), >14 drinks per week). The risk of MetS is higher in men who consume more than 7 drinks a week with high hsCRP levels than that in nondrinkers. Study is well designed, but I recommend some changes to improve quality of this research: 

1) Please add approval of Ethic committee and data about Inform consent.

Answer (line 432 to 436)

We have corrected what you pointed out such as:

Informed consent was waived because re­searchers only accessed and analyzed the secondary data set. The authors assert that all procedures contributing to this work com­plied with the ethical standards of the relevant national and institu­tional committees on human experimentation and with the Hel­sinki Declaration of 1975, as revised in 2008.

2) Table 1 and table 2: please add rows with data about occurence of MetS, and data about occurence of MetS with 3, 4 and 5 individual components of MetS.

Answer

We added the prevalence of metabolic syndrome in Tables 1 and 2 (3 after revision). Due to the sample size problem, the only case with 3 or more components of the metabolic syndrome was described.

3) Please add  the separate table comparing patients with and without MetS.

Answer (Table 2)

As you pointed out, we have created a new table 2.

4) Please add the separate table comparing only men with high hsCRP, add 4 collumns ND, LM, HM, HD, please include data about occurence of MetS, and data about occurence of MetS with 3, 4 and 5 individual components of MetS.

Answer (Table 5)

As you pointed out, we have created a new table 5.

5) Please add separate table with data from univariate analysis.

Answer

Unfortunately, we didn't understand exactly what you pointed out, so we couldn't create a table.

6) Part Discussion: Please add chapter with data about association about alcohol and subclinical inflammation

Answer (line 383 to 389)

We did not understand exactly what you pointed out. However, we added something about the newly created table 5 to the discussion.

7) Conclusion: Change term individuals for men.

Answer

Line 40, 302, and 319, we have corrected what you pointed out.

My final decision is major revision. 

Round 2

Reviewer 3 Report

Dear sir,

i recommend to publish an article Association of High-sensitivity C-reactive Protein and Alcohol Consumption on Metabolic Syndrome in Korean Men.